# Needle Tract Seeding after Endoscopic Ultrasound Tissue Acquisition of Pancreatic Lesions: A Systematic Review and Meta-Analysis

**DOI:** 10.3390/diagnostics12092113

**Published:** 2022-08-31

**Authors:** Antonio Facciorusso, Stefano Francesco Crinò, Paraskevas Gkolfakis, Daryl Ramai, Benedetto Mangiavillano, Juliana Londoño Castillo, Saurabh Chandan, Babu P. Mohan, Francesca D’Errico, Francesco Decembrino, Viktor Domislovic, Andrea Anderloni

**Affiliations:** 1Gastroenterology Unit, Department of Medical and Surgical Sciences, University of Foggia, 71122 Foggia, Italy; 2Gastroenterology and Digestive Endoscopy Unit, Department of Medicine, The Pancreas Institute, University Hospital of Verona, 37134 Verona, Italy; 3Department of Gastroenterology, Hepatopancreatology, and Digestive Oncology, CUB Erasme Hospital, Université Libre de Bruxelles (ULB), 1050 Brussels, Belgium; 4Gastroenterology and Hepatology, University of Utah Health, Salt Lake City, UT 84112, USA; 5Gastrointestinal Endoscopy Unit, Humanitas—Mater Domini, 21053 Castellanza, Italy; 6Gastroenterology Unit, CHI Health Creighton University Medical Center, Omaha, NE 68131, USA; 7UOC Gastroenterologia ed Endoscopia Digestiva, Ente Ecclesiastico—Ospedale Generale Regionale “F.Miulli”, 70021 Acquaviva delle Fonti, Italy; 8Department of Gastroenterology and Hepatology, University Hospital Centre Zagreb, 10000 Zagreb, Croatia; 9Gastroenterology and Digestive Endoscopy Unit, Fondazione I.R.C.C.S. Policlinico San Matteo, 27100 Pavia, Italy

**Keywords:** FNA, EUS, FNB, cancer, tumor

## Abstract

There is limited evidence on the incidence of needle tract seeding (NTS) in patients undergoing endoscopic ultrasound (EUS) tissue acquisition (TA) of pancreatic lesions. This meta-analysis aimed to assess the incidence of NTS after EUS-TA. With a search of the literature up until April 2022, we identified 10 studies (13,238 patients) assessing NTS incidences in patients undergoing EUS-TA. The primary outcome was NTS incidence. The secondary outcome was a comparison in terms of peritoneal carcinomatosis incidence between patients who underwent EUS-TA and non-sampled patients. Results were expressed as pooled rates or odds ratio (OR) and 95% confidence intervals (CI). The pooled rate of NTS was 0.3% (95% CI 0.2–0.4%), with no evidence of heterogeneity (I^2^ = 0%). Subgroup analysis based on the type of sampled lesion confirmed this finding both in patients with pancreatic adenocarcinoma (0.4%, 0.2–0.6%) and in patients with cystic pancreatic lesions (0.3%, 0.1–0.5%). No difference in terms of metachronous peritoneal dissemination was observed between patients who underwent EUS-TA and non-sampled patients (OR 1.02, 0.72–1.46; *p* = 0.31), with evidence of low heterogeneity (I^2^ = 16%). Rates of NTS after EUS-TA are very low; therefore, EUS-TA could be safely performed in a pre-operative setting.

## 1. Introduction

Endoscopic ultrasound (EUS) represents a fundamental diagnostic tool for the management of pancreatic lesions. However, simple morphological evaluations are not sufficient for the definitive characterization of pancreatic lesions; hence, EUS-guided tissue acquisition (TA) for cytopathological and histological diagnosis by means of fine needle aspiration (FNA) and, more recently, fine needle biopsy (FNB) is usually needed [1,2].

While EUS-FNA may still play a competitive role as compared to EUS-FNB when rapid-on-site cytological evaluation (ROSE) is available, EUS-FNB was found to overperform standard FNA for tissue sampling of several solid lesions in the absence of a pathologist in the endoscopic room [3,4,5,6]. Moreover, in spite of the favorable diagnostic outcomes of EUS through-the-needle biopsy (TTNB) and confocal laser endomicroscopy (CLE), EUS-FNA is still frequently used in the diagnostic algorithm of cystic pancreatic lesions (PCLs) [7,8,9]. Although uncommon, one potential and serious complication of EUS-TA is needle tract seeding (NTS) [10]. NTS represents the iatrogenic implantation of tumoral cells along the needle tract, and it has been described after EUS-TA of both solid lesions and pancreatic cysts [11,12]. Consequences of NTS may appear even several months after EUS-TA as the occurrence of gastric wall nodules or peritoneal carcinomatosis [10]. Given the potential negative effect of NTS on prognosis, the importance of preoperative EUS-FNA for the diagnosis of resectable pancreatic lesions might be questioned, and the exact definition of the incidence of this fearful complication may reappraise the use of EUS-TA in this setting.

Although several meta-analyses assessing the adverse event rates after EUS-TA have been recently published, a comprehensive systematic review focused on the incidence of NTS in patients undergoing EUS-FNA/B is still lacking [13,14]. Therefore, we performed a meta-analysis to provide a pooled estimate of the incidence of NTS after EUS-TA. Moreover, we assessed the comparison in terms of peritoneal carcinomatosis occurrence between patients who underwent EUS-FNA/B versus patients not submitted to EUS-TA.

## 2. Methods

### 2.1. Selection Criteria

The literature search strategy was based on the following inclusion criteria: (1) observational or cohort studies assessing NTS in adult patients undergoing EUS-TA for resectable pancreatic lesions; (2) studies published in English; (3) studies with at least 1 year of follow-up. Small case series < 10 patients, non-endoscopic studies, review articles, and animal models were excluded.

### 2.2. Search Strategy 

A literature search was conducted on PubMed, EMBASE, Cochrane Library, and Google Scholar, including all studies fulfilling the inclusion criteria published up until April 2022, based on the string ((((endoscopic ultrasound) OR (EUS)) AND (FNA)) OR (FNB)) AND (seeding). Relevant reviews and meta-analyses in the field were examined for potential additional suitable studies. Authors of included studies were contacted to obtain full text or further information when needed. A manual search on the proceedings of the main international endoscopic and gastroenterological conferences was also performed. Data extraction was performed by 2 authors (AF and SFC), and the quality of included studies was rated by 2reviewers independently (AF and PG) based on the Newcastle-Ottawa scale for non-randomized studies [15]. Disagreements were solved by discussion and after a third opinion (AA).

### 2.3. Outcomes Assessed

The primary outcome was NTS incidence, defined as the occurrence of a metachronous lesion appearing in the needle tract. The secondary outcome was the comparison between patients who underwent EUS-TA vs. patients who did not undergo EUS-TA in terms of peritoneal tumoral dissemination during follow-up. Peritoneal dissemination was defined as the development of ascites with malignant cytology or pathologic confirmation of malignancy in peritoneal/omental/mesenteric tumor implants or cross-sectional image findings indicative of carcinomatosis. 

### 2.4. Statistical Analysis

Study outcomes were pooled by using a random-effects model based on the DerSimonian and Laird test, and results were expressed as rates and a 95% confidence interval (CI). The random-effects method incorporates an assumption that the different studies are estimating different yet related intervention effects. The method is based on the inverse-variance approach, making an adjustment to the study weights according to the extent of variation, or heterogeneity, among the varying intervention effects. The random-effects method and the fixed-effect method produce identical results when there is no heterogeneity among the studies. Where there is heterogeneity, confidence intervals for the average intervention effect are wider if the random-effects method is used rather than a fixed-effect method, and if corresponding claims of statistical significance are more conservative. It is also possible that the central estimate of the intervention effect would change if there were relationships between the observed intervention effects and sample sizes. Therefore, the comparison between the two groups was based on a random-effects model, and the results were expressed as an odds ratio (OR) and a 95% CI. The presence of heterogeneity was calculated using I^2^ tests with I^2^ < 20% and interpreted as low-level and an I^2^ between 20% to 50% interpreted as moderate heterogeneity. Any potential publication bias was verified by using a visual assessment of funnel plots. A funnel plot is a simple scatter plot of the intervention effect estimates from individual studies against some measure of each study’s size or precision. In common with forest plots, it is most common to plot the effect estimates on the horizontal scale, and, thus, the measure of study size on the vertical axis. This is the opposite of conventional graphical displays for scatter plots, in which the outcome (e.g., intervention effect) is plotted on the vertical axis and the covariate (e.g., study size) is plotted on the horizontal axis. The name ‘funnel plot’ arises from the fact that the precision of the estimated intervention effect increases as the size of the study increases. Effect estimates from small studies will therefore scatter more widely at the bottom of the graph, with the spread narrowing among larger studies. In the absence of bias, the plot should approximately resemble a symmetrical (inverted) funnel. If there is bias, for example, because smaller studies without statistically significant effects remain unpublished, this will lead to an asymmetrical appearance of the funnel plot with a gap at the bottom corner of the graph. In this situation, the effect calculated in a meta-analysis will tend to overestimate the intervention effect. The more pronounced the asymmetry, the more likely it is that the amount of bias will be substantial. Sensitivity analysis was conducted restricted to (1) the type of sampled lesion (pancreatic adenocarcinoma (PDAC) versus PCLs) and (2) according to study location (East versus West). All statistical analyses were conducted using RevMan (version 5.0 for Windows; the Cochrane Collaboration, Oxford, UK), OpenMeta [Analyst] software, and R 3.0.2 (R Foundation for Statistical Computing, Vienna, Austria). For all calculations, a 2-tailed *p* value of less than 0.05 was considered statistically significant.

## 3. Results

### 3.1. Studies

As shown in Figure 1, of 89 studies initially identified, after the exclusion of articles not fulfilling the inclusion criteria and of 2 duplicate studies [16,17], 10 studies [18,19,20,21,22,23,24,25,26,27] with 13,238 patients were included in the meta-analysis. Of 10 included studies, 8 were retrospective case–control studies, of which 6 compared EUS-TA vs. non-sampling [19,20,22,24,26,27], 1 study compared EUS-TA vs. percutaneous FNA [21], and 1 study compared EUS-TA vs. ERCP [23]. The other two included studies were a nationwide survey [18] and a retrospective single cohort study [25]. The main characteristics of the included studies were reported in Table 1. The recruitment period ranged from 1995 to 2018. Seven studies were conducted in Asia [18,19,23,24,25,26,27] and three studies in the USA [20,21,22]. The majority of treated patients were male, and the mean lesion size ranged from 2.3 to 4.7 cm. Seven studies included only patients with resectable solid lesions [18,19,21,23,24,25,27]; two studies included both patients with solid lesions and resectable PCLs [20,26], and one study included only patients with IPMN with indication of surgery [22]. The vast majority of sampled lesions were PDAC. Median follow-up length ranged from 599 days to 56.9 months. Quality was deemed moderate to high in seven studies [18,19,20,22,24,26,27], whereas three studies [21,23,25] were rated as low-quality articles, mainly due to incomplete outcome reporting. Details on the quality assessment of the included articles are shown in Appendix A.

### 3.2. NTS

As depicted in Figure 2, the pooled rate of NTS was 0.3% (95% CI 0.2–0.4%), with no evidence of heterogeneity (I^2^ = 0%). Given the preponderant weight of the study by Kitano et al. [18], we performed a leave-one-out analysis excluding this study. As reported in Appendix A, this analysis confirmed the aforementioned results with a pooled NTS rate of 0.5% (0.1–0.9%). Subgroup analysis based on the type of sampled lesion confirmed this finding both in patients with PDAC (0.4%, 0.2–0.6%; Table 2 and Appendix A) and in patients with PCLs (0.3%, 0.1–0.5%; Table 2 and Appendix A). Again, no evidence of heterogeneity was observed in any of the subgroups analyzed, and no evidence of publication bias was detected by a visual inspection of the funnel plot (Figure 3). Overall, out of 44 cases of NTS reported, 39 occurred after sampling of the head of pancreas lesions, and 5 occurred after sampling distal lesions. 

### 3.3. Peritoneal Dissemination

A comparison between EUS-TA vs. non-sampling was reported in Figure 4. Based on eight studies [19,20,22,23,24,25,26,27], no difference in terms of metachronous peritoneal dissemination was observed between the two groups (OR 1.02, 0.72–1.46; *p* = 0.31), with evidence of low heterogeneity (I^2^ = 16%). As depicted in Appendix A, no evidence of publication bias was observed. Figure 1 shows the exclusion of articles not fulfilling the inclusion criteria of the 89 studies initially identified.

## 4. Discussion

EUS-TA plays a pivotal role in the diagnostic management of patients with both pancreatic masses and PCLs. In fact, pathological confirmation is fundamental for the diagnosis and avoids unnecessary surgery for benign lesions [1,2]. Histological confirmation with EUS-TA is also required prior to neoadjuvant chemotherapy and seems to improve survival outcomes even in the presence of resectable masses [28]. Although low rates of adverse events have been described after EUS-TA, NTS represents a dreadful complication that may determine a poorer prognosis in these patients [10]. NTS has been reported in several case reports [11,29], but its real incidence is still unknown. To the best of our knowledge, this is the first meta-analysis aimed at systematically reviewing the incidence of NTS in patients who underwent EUS-TA for pancreatic lesions. 

By conducting a meta-analysis of 10 studies, we made several key observations. First, the pooled rate of NTS is 0.3% (95% CI 0.2–0.4%), with no difference between patients with solid masses (0.4%, 0.2–0.6%) and patients with PCLs (0.3%, 0.1–0.5%). The vast majority of the NTS cases appeared as gastric submucosal tumor-like masses, and pathological analyses showed that tumor cells were present in the submucosal layers of the gastric wall. The detection of the seeding neoplasia was commonly obtained using upper gastrointestinal endoscopy and other imaging modalities such as positron emission tomography (PET). Although a specific survival analysis was not feasible due to the lack of data, some studies did not report a significant decrease in the overall survival after early detection and the consequent resection of NTS tumor [23,27]. In general, EUS-TA has been found to not influence overall survival in patients with resectable PDAC, thus meaning that even eventual unrecognized NTS events are not able to lead to a poorer prognosis in these patients [20,23,30]. As a consequence, EUS-FNA/B should not be discouraged when there is a suspicion of pancreatic cancer. Patients should be followed-up carefully over time as NTS might appear years after EUS-TA; hence, it may be difficult to predict when NTS occurs [10]. 

The second key observation of our study was the comparable incidence of metachronous peritoneal carcinomatosis between patients with pancreatic lesions submitted to EUS-TA and non-sampled patients. A retrospective analysis found a lower intraoperative detection of peritoneal carcinomatosis in pancreatic cancer patients who underwent pre-operative EUS-TA compared to percutaneous FNA [21]. However, because the initial laparoscopic staging was not performed in all cases before neoadjuvant therapy, a link between peritoneal carcinomatosis and tumor seeding following FNA could be postulated, although only indirectly. 

Although peritoneal carcinomatosis may represent a natural evolution of the clinical course of pancreatic cancer patients, the similar rates of carcinomatosis during the follow-up speak against an increased risk of peritoneal tumoral dissemination due to EUS-TA. 

One may argue that follow-up lengths were quite heterogeneous among the included studies. In fact, as previously commented, NTS may develop even years after EUS-TA; therefore, the incidence of NTS reported in these studies might have been underestimated. However, PDAC represents an aggressive disease with limited disease-free survival and overall survival, and we include only studies with at least 1 year of follow-up (median follow-up of 23 months), a period of time that could be considered sufficient to detect eventual NTS events. In fact, 1 year also represents the timespan for clinical observation suggested by the American Gastroenterology Association to confirm the diagnosis of PDAC after EUS-TA in non-resected patients [31]. 

Usually, NTS is described after trans-gastric sampling of distal lesions, while only a few anecdotal cases have been reported following trans-duodenal puncture of the head of pancreas lesions. In fact, gastric implantation, the common appearance of NTS, can occur when EUS-TA is performed via the stomach, targeting cancer of the pancreatic body or tail. On the other hand, in cancers of the pancreatic head, EUS-FNA/B is usually performed via the duodenal bulb; therefore, the puncture site falls within the surgically resected area.

Preliminary molecular studies seem to suggest a potentially higher NTS rate than previously described. A prospective clinical study conducted by Levy et al. found an unexpectedly high presence of malignant cells by means of luminal fluid cytology in the gastrointestinal luminal fluid in patients who underwent diagnostic EUS-FNA for pancreatic cancer [32]. However, further studies are needed in this setting to establish the exact rate of tumoral cell translocation due to EUS-TA, although the prognostic impact of these events is unlikely to be significant, as previously commented [33]. 

Because of the limited number of reported cases of NTS, it remains unknown which procedural or lesion-related factors might be significantly correlated with the occurrence of NTS. Sakamoto et al. conducted an experiment using an agar model, and they considered that the slow-pull technique and the use of a puncture needle with a side hole might result in NTS [34]. Although it is unclear how much these results could apply to clinical practice, it also gives us a warning, and this aspect should be carefully considered. 

Several strategies have been proposed to reduce tumor seedings. Yane et al. proposed that EUS-FNA should only be used in patients requiring a pathological diagnosis to develop more accurate treatment strategies (for example, patients with pancreatic cancer who should undergo preoperative neoadjuvant therapy) or in patients in which the disease is difficult to diagnose by imaging [17]. In addition, they proposed that the distance between the scope and the target lesion should be as short as possible. Tomonari et al. [35] proposed that if the surgical resection does not include puncture needle tracts or puncture results cannot change treatment options, EUS-guided sampling should be avoided, or the number of samplings should be as low as possible. Therefore, one may consider replacing EUS-FNA with EUS-FNB to improve diagnosis with a more limited number of passes, thus decreasing the risk of NTS. Finally, several authors suggest performing a regular detection of blood tumor markers, imaging, and endoscopy, even when the seeding tract is radically resected.

There are certain limitations to this study. First, the low number of included studies and enrolled patients requires particular caution in interpreting our findings. However, we deliberately decided to restrict inclusion criteria to a series with at least 10 patients, thus excluding small case reports/series, to provide more robust and homogenous outcome estimates. Moreover, as previously stated, we included only studies with at least 1 year of follow-up in order to properly detect NTS events. Of note, follow-up lengths were significantly longer in the two studies that included PCLs, because in that case, tumor recurrence/seeding may occur even several years after resection [20,22]. Second, the limited number of comparative studies and the lack of randomized trials prevented a robust assessment of the direct comparison between the two groups in terms of peritoneal carcinomatosis, which represents a potentially indirect consequence of EUS-TA. Moreover, the lack of data did not enable us to perform subgroup analyses based on specific technical characteristics such as the number of needle passes, the location of the sampled lesion, and the type of needle used. Furthermore, the type of resection (whether R0 or R1) could not be analyzed as a predictor of NTS due to the lack of data. Third, survival outcomes could not be compared between the two groups due to the lack of data. Therefore, definitive assumptions on the potential prognostic impact of NTS events could not be drawn, although this did not seem to be significant in previous reports [20,23,30]. Finally, a subgroup assessment based on the type of needle used (whether FNB vs. FNA or within the FNB group between end-cutting vs. side-fenestrated needles) could not be performed, again, due to the lack of data. In fact, NTS represents a rare event, and only very large nationwide studies can be properly powered to detect the incidence of tumoral seeding in specific subsets of patients. 

## 5. Conclusions

In conclusion, this meta-analysis highlights that NTS represents an uncommon complication of EUS-TA, and it is very unlikely to significantly affect patient outcomes. Therefore, EUS-TA of pancreatic lesions should not be discouraged in the pre-operative setting. Authors should discuss the results and how they can be interpreted from the perspective of previous studies and the working hypotheses. The findings and their implications should be discussed in the broadest context possible. Future research directions may also be highlighted.

It has been reported that EUS-FNA for pancreatic tumors has high sensitivity and specificity; however, this procedure is not devoid of complications such as bleeding, pancreatitis, and post-procedural pain, although, due to their low incidence, it is considered a safe procedure. However, as demonstrated in our meta-analysis, the risk of NTS should not be overlooked. The first case of NTS after EUS-FNA in a patient with invasive ductal carcinoma derived from IPMN was reported in 2003 [11], and then in 2005, NTS after EUS-FNA was reported in a patient with pancreatic ductal adenocarcinoma [29]. 

Usually, the puncture route is included in the resection range for pancreatic head cancer. Therefore, in these cases, NTS can be detected during surgery. Hence, intraoperative assessment for gastric wall metastasis is important, as well as postoperative assessment, and if the surgeon suspects gastric wall metastasis intraoperatively, partial gastrectomy should be performed without hesitation. 

The treatment policy for pancreatic cancer varies according to the tumor resectability; surgery is the first treatment choice for resectable pancreatic cancer. For borderline resectable pancreatic cancer, it is a dominant opinion that neoadjuvant chemoradiotherapy is known to improve the prognosis, and for unresectable cases, chemotherapy is chosen. If chemotherapy is chosen for pancreatic cancer, including preoperative treatment, it is necessary to differentiate it from other pancreatic tumors due to NTS. 

When EUS-FNA is performed for pancreatic body or tail cancer, which is not included in the resection range, we should be aware of the risk of developing NTS in the gastric wall. In order to avoid this dreadful complication, a biopsy needle with a covering sheath should be used. 

## Figures and Tables

**Figure 1 diagnostics-12-02113-f001:**
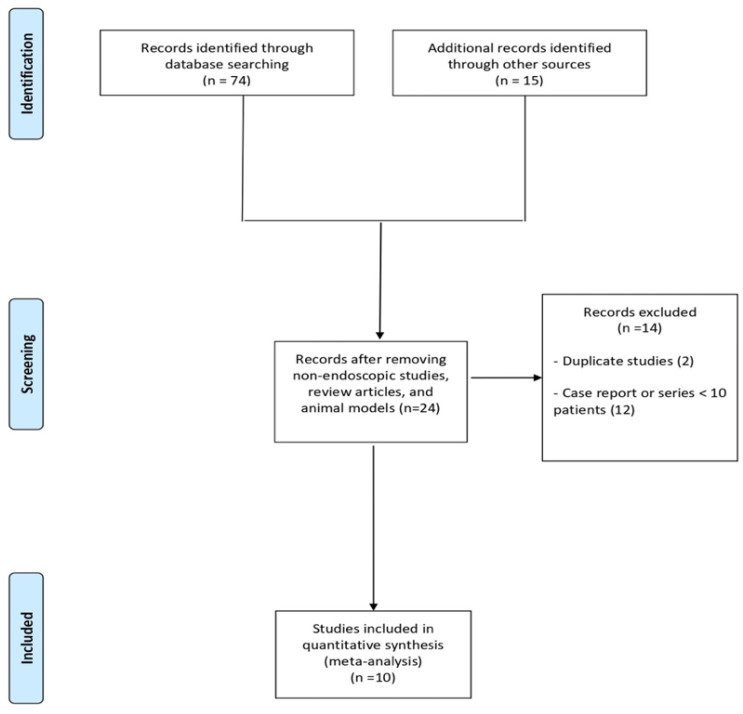
Flow chart of the included studies.

**Figure 2 diagnostics-12-02113-f002:**
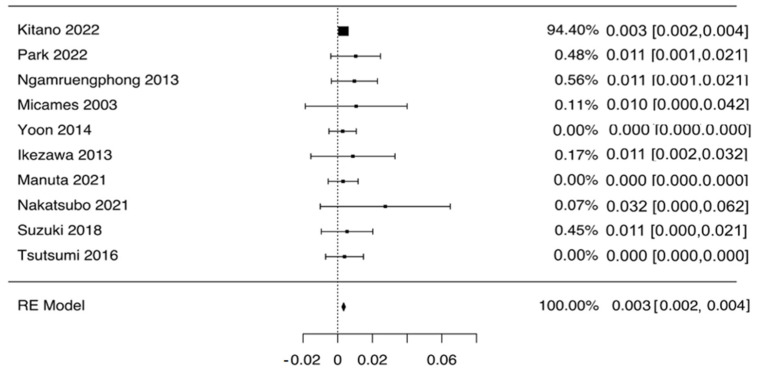
Pooled analysis of needle tract seeding rate in patients who underwent EUS-guided tissue acquisition. Pooled rate of needle tract seeding was 0.3% (95% CI 0.2–0.4%), with no evidence of heterogeneity (I^2^ = 0%).

**Figure 3 diagnostics-12-02113-f003:**
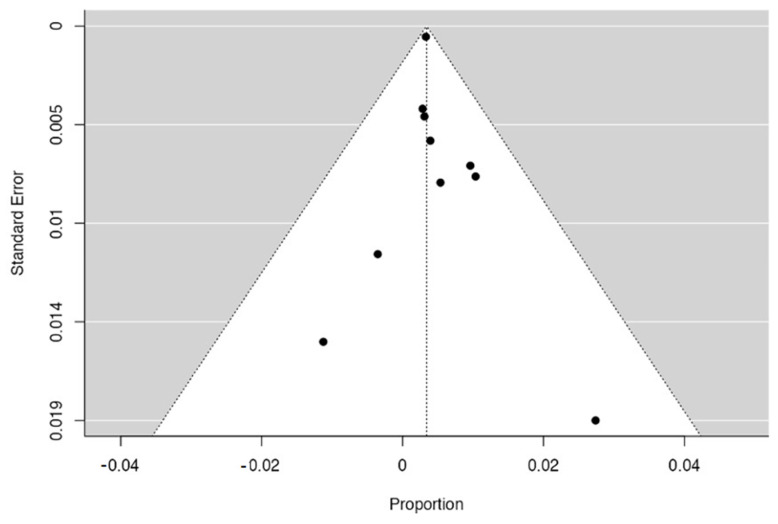
Funnel plot concerning the risk of publication bias.

**Figure 4 diagnostics-12-02113-f004:**
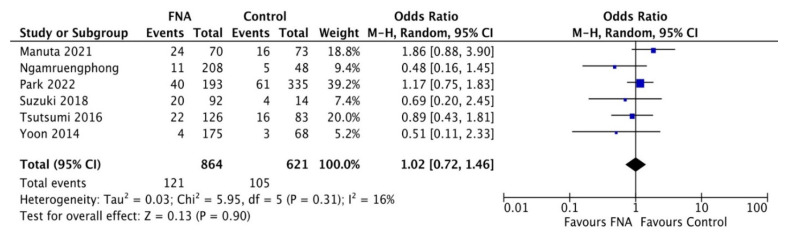
Forest plot comparing EUS-guided tissue acquisition versus non-sampling in terms of metachronous peritoneal dissemination. No difference in terms of metachronous peritoneal dissemination was observed between the two groups (OR 1.02, 0.72–1.46; *p* = 0.31), with evidence of low heterogeneity (I^2^ = 16%).

**Table 1 diagnostics-12-02113-t001:** Characteristics of included studies.

Study	Country	Sample Size; Sampling Technique/Kind of Lesion (PDAC)	Study Period/ Design	Control Group (Sample Size)	Age Years	Gender Male	Lesion Size (cm)/Location Head of Pancreas	Inclusion Criteria	Follow-Up Length/R0 Resection
Kitano 2022 [18]	Japan	12,109; FNB/FNA/ 9300 (76.8%)	2010–2018/ Nationwide survey	NR	NR	NR	NR/ NR	Resectable solid lesions	NR/ NR
Park 2022 [19]	Korea	193; FNA/FNB/ 193 (100%)	2007–2017/ Retrospective case–control	Patients not submitted to EUS-TA (335)	61.4 ± 10.6	115 (59.6%)	3.4 ± 1.8/ 0 (0%)	Resectable distal PDAC	34.1 months (31.7–36.4)/ 129 (66.8%)
Ngamruengphong 2013 [20]	USA	208; FNA/ 174 (83.7%)	1996–2012/ Retrospective case–control	Patients not submitted to EUS-TA (48)	66 ± 11	105 (50.5%)	4.7 ± 11.7/ 146 (70.2%)	Resectable solid and cystic lesions	23 months (0–111)/ 182 (87.5%)
Micames 2003 [21]	USA	46; FNA/ 46 (100%)	1995–2001/ Retrospective case–control	Patients submitted to percutaneous FNA (43)	61 (42–83)	23 (50%)	3 ± 1/ 41 (89%)	PDAC	NR/ NR
Yoon 2014 [22]	USA	175; FNA/ 0 (0%)	1999–2010/ Retrospective case–control	Patients not submitted to EUS-TA (68)	68 (39–92)	75 (42.8%)	35 pts > 3.5/ 102 (58.3%)	IPMN with indication to surgery	56.9 months (6–136.6)/ NR
Ikezawa 2013 [23]	Japan	56; FNA/ 56 (100%)	2006–2009/ Retrospective case–control	Patients submitted to ERCP (161)	64.2 ± 9.1	35 (62.5%)	2.8 ± 1.4/ 30 (53.6%)	PDAC	599 days ± 426/ NR
Maruta 2021 [24]	Japan	160; FNA/ 160 (100%)	2005–2017/ Retrospective case–control	Patients not submitted to EUS-TA (73)	70 (46–85)	100 (62.5%)	2.3 (0.8–5.3)/ 115 (71.9%)	Resectable PDAC	NR/ NR
Nakatsubo 2021 [25]	Japan	73; FNB/ 67 (91.7%)	2014–2016/ Retrospective	No	67.2 ± 10.2	45 (61.6%)	3 (0.7–14)/ 40 (54.8%)	Resectable solid masses	NR 45 (67.2%)
Suzuki 2018 [26]	Japan	92; FNA/ 84 (91.3%)	2006–2016/ Retrospective case–control	Patients not submitted to EUS-TA (14)	71.5 (48–86)	54 (58.7%)	NR/ NR	Resectable IPMN or PDAC	NR/ NR
Tsutsumi 2016 [27]	Japan	126; FNA/ 126 (100%)	1996–2012/ Retrospective case–control	Patients not submitted to EUS-TA (83)	66.6 ± 8.9	73 (57.9%)	2.4 ± 1.2/ 85 (67.4%)	Resectable PDAC	>1 year/ NR

Continuous data are presented as mean ± standard deviation or median (range), and categorical data are presented as absolute numbers (percentage). Abbreviations: EUS-TA, endoscopic ultrasound tissue acquisition; IPMN, intraductal pancreatic mucinous neoplasia; FNA, fine-needle aspiration; FNB, fine-needle biopsy; PDAC, pancreatic ductal adenocarcinoma.

**Table 2 diagnostics-12-02113-t002:** Sensitivity analysis of needle tract seeding. Sensitivity analysis was performed based on (a) sampled lesion (adenocarcinoma vs. pancreatic cysts), and (b) study location (East vs. West).

Variable	Subgroup	No. of Cohorts	No. of Patients	Summary Estimate (95% CI)	Within-Group Heterogeneity (I^2^)
Treatment Success					
Sampled lesion	PDAC	9	10,206	0.4% (0.2–0.6%)	0%
	PCL	3	187	0.3% (0.1–0.5%)	0%
Study location	East	7	12,809	0.3% (0.1–0.7%)	0%
	West	3	429	0.3% (0.2–0.5%)	0%

Abbreviation: CI, Confidence Interval.

## Data Availability

Not applicable.

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
