# Peer review of "Needle Tract Seeding after Endoscopic Ultrasound Tissue Acquisition of Pancreatic Lesions: A Systematic Review and Meta-Analysis"

_diagnostics, 2022, doi:10.3390/diagnostics12092113_

Round 1

Reviewer 1 Report

Dear author,
I had the pleasure of rewiving your study.

However, where possible, I think it would be necessary to highlight the strong points and justify some limitations. Especially:

For the included studies, where indicated, it should be specified:
- the location of the primary tumor (cephalic vs distal).
In the first case the biopsy is trans duodenal. For distal locations the biopsy is performed transgastrically with an increased risk of NTS.
- the percentage of patients who have had curative surgery and in particular the percentage of incomplete resection. In case of resection R1 increased risk of loco-regional recurrence.
- for studies that take into consideration NTS and PCLs the percentage of adenocarcinoma.

For information not detailed, it might be useful to add a paragraph "limitations of the study".

Kind regards

Author Response

Dear author,
I had the pleasure of rewiving your study.
However, where possible, I think it would be necessary to highlight the strong points and justify some limitations. Especially:

For the included studies, where indicated, it should be specified:
- the location of the primary tumor (cephalic vs distal).
In the first case the biopsy is trans duodenal. For distal locations the biopsy is performed transgastrically with an increased risk of NTS.

RE: We really thank the reviewer for this important observation. We added to Table 1 the location of the sampled lesion and in the text (Results section) we reported how many cases of NTS occurred after sampling of head of pancreas lesions and how many after sampling of distal lesions. However, due to the limited number of studies, a specific subgroup analysis based on this parameter was unfeasible and this aspect was reported among the limitations to the study.

- the percentage of patients who have had curative surgery and in particular the percentage of incomplete resection. In case of resection R1 increased risk of loco-regional recurrence.

RE: This information was added (when available) to Table 1 and this aspect was commented among the limitations to the paper. Thank you!

- for studies that take into consideration NTS and PCLs the percentage of adenocarcinoma.

RE: This information was reported in Table 1 and we commented in the text (Results section)  that the vast majority of sampled lesions were PDAC.

For information not detailed, it might be useful to add a paragraph "limitations of the study".
Kind regards

RE: As reported above, we commented the missing informations among the limitations to the study. Thank you!

Reviewer 2 Report

This is a systematic review and meta-analysis of needle tract seeding after EUS-guided tissue acquisition.

1. Was the term "NTS" used in all studies? For example, Yoon et al. did not use NTS and peritoneal seeding. Peritoneal seeding does not necessarily mean NTS. How did the authors collect data on NTS and peritoneal tumoral dissemination in this study by Yoon.  In general, the term "NTS" means tumor arising in the needle tract such as the gastric wall, and I am not sure it would be seen after EUS-FNA for pancreatic cystic lesions. Please define NTS and peritoneal dissemination.

2.  Conclusion. Line 306. "the incidence rate is as low as 1.03%." Where was this number of 1.03% shown in the text?

3.  Conclusion. Line 309. "the common type of pancreatic adenocarcinoma" "pancreatic ductal adenocarcinoma" might be appropriate here.

Author Response

This is a systematic review and meta-analysis of needle tract seeding after EUS-guided tissue acquisition.

  1. Was the term "NTS" used in all studies? For example, Yoon et al. did not use NTS and peritoneal seeding. Peritoneal seeding does not necessarily mean NTS. How did the authors collect data on NTS and peritoneal tumoral dissemination in this study by Yoon.  In general, the term "NTS" means tumor arising in the needle tract such as the gastric wall, and I am not sure it would be seen after EUS-FNA for pancreatic cystic lesions. Please define NTS and peritoneal dissemination.

RE: The definition of NTS (an acronym used in all the studies reporting this event, the Study by Yoon did not report this event) was consistent across the included studies and it was defined as “the occurrence of a metachronous lesion appearing in the needle tract”, as reported in the text at page 2. Based on the meaningful reviewer’s suggestion, we added to the same chapter the definition of peritoneal dissemination. Specifically, the study by Yoon et al did not report any cases of NTS and in fact the incidence was 0% in the forest plot depicted in Figure 2.

  1. Line 306. "the incidence rate is as low as 1.03%." Where was this number of 1.03% shown in the text?

RE: This was a typo and it was amended in the text. Thank you!

  1. Line 309. "the common type of pancreatic adenocarcinoma" "pancreatic ductal adenocarcinoma" might be appropriate here.

RE: The sentence was amended as suggested. Thank you!

Round 2

Reviewer 1 Report

Dear Dr. Facciorusso,
thank you for making the requested changes.

Kind rergards